# The Influence of Consumer Purchases on Purchase-Related Happiness: A Serial Mediation of Commitment and Selective Information Processing

**DOI:** 10.3390/bs13050396

**Published:** 2023-05-10

**Authors:** Dongyoup Kim, Yeosun Yoon

**Affiliations:** 1College of Business, Gachon University, Seongnam 13120, Republic of Korea; 2KAIST College of Business, Seoul 02455, Republic of Korea; yyoon@kaist.ac.kr

**Keywords:** selective information processing, material purchase, experiential purchase, commitment to-purchase, purchase-related happiness

## Abstract

In the literature on material and experiential purchases, it has consistently been demonstrated that the relationship between the consumer’s purchase type and purchase-related happiness favors experiential purchases. This research aims to extend the literature by examining how experiential purchases lead to greater purchase-related happiness due to the individual’s processing of external information, especially in the online review context. An experiment was conducted to show that experiential purchases lead to greater commitment to decisions and a higher relative reliance on positive reviews (as opposed to negative reviews) than material purchases. The results of a serial mediation test indicate that such differences lead to greater purchase-related happiness. Based on these findings, we can deepen our understanding of the relationship between purchase type and purchase-related happiness from the perspective of information processing.

## 1. Introduction

As the use of the Internet and electronic commerce continues to grow rapidly, individuals are gaining more opportunities to share their knowledge and opinions about their purchases [1]. The development of the Internet and electronic commerce has also changed the way people make purchases. Consumers can now easily access the market and search for a wide range of information as they need [2,3]. Online shopping sites provide information about not only the products but also commentaries left by individuals who have already made the purchase. For example, over 92 percent of consumers consult product reviews prior to making a purchase decision, and these reviews serve as a dependable sales assistant, contributing to the growth in sales revenue [4]. These commentaries contain both positive and negative opinions about the products, which subsequently influences attitude formation before the purchase [5]. Moreover, information from other sources, such as posts and commentaries related to social media influencers, also effectively contributes to consumers’ purchase decision-making [6,7]. Even after the purchase, consumers often use the reviews of other consumers to evaluate whether their purchases were the right choice and to assess satisfaction with the purchase [8,9]. Based on the analysis of data collected from product evaluation forums, it was discovered that 38.8% of reviews were acquired after the purchase, rather than prior to it [10]. For individual consumers, the experiences of other consumers are perceived as the most reliable source of information; thus, they adjust their purchase-related happiness according to the influential information [11].

The best way to induce greater purchase-related happiness is to expose individuals only to positive commentaries, although this is legally prohibited. Therefore, companies must find ways to make people pay more attention to positive messages than negative ones. Studies have explored the factors that encourage selective information processing, such as individuals in a negative [12] or anxious mood state [13,14], those who face congenial information [15], those with a high information load [16,17,18], or those highly committed to their initial attitudes [19,20]. However, the literature on selective information processing has not paid much attention to situational or contextual effects, other than the influence of individual psychological variables. Therefore, this research aims to add to the existing literature by investigating the effect of the type of purchase individuals make.

In this research, we focus on the purchase type dichotomy of material and experiential purchases. Depending on whether the purchase is intended for material possession or a life experience, individuals can either make material purchases or experiential purchases [21,22]. According to the literature on material and experiential purchases, there is a consistent difference in purchase-related happiness between the two types of purchases, with experiential purchases providing greater and longer-lasting happiness than material purchases [23,24,25,26]. This superiority of experiential purchases is attributed to past purchases, evaluation stages, and anticipated purchases [8,27]. However, existing studies have a limitation when it comes to understanding how external information is processed and used in experiential purchases. While it is known that consumers of experiential purchases are more likely to communicate about their purchases than those of material purchases [28], it is not clear how external information is used and utilized. As external information is difficult for managers to directly control, practical implications of this knowledge can be of great importance.

Consequently, we assert that experiential purchases induce greater purchase-related happiness compared to material purchases because individuals who make experiential purchases tend to rely more heavily on positive information. Specifically, we propose that the relationship between purchase type and purchase-related happiness is mediated by commitment-to-purchase and selective reliance on positive information. Additionally, we demonstrate the proposed underlying mechanism by illustrating that trustworthy feedback allows individuals to place greater emphasis on positive (vs. negative) information, even when it comes to material purchases. 

According to the above findings, this study makes the following two contributions to the existing literature. First, it extends the existing body of literature in the context of selective information processing by suggesting that what individuals purchase can be an antecedent of their tendency to process information discriminately. Second, we examine the underlying process of why an experiential purchase is superior in terms of purchase-related happiness from the perspective of information processing, which is a novel approach compared to previous research that has focused on internal sources of information such as the perceived right choice [8], self-relatedness [24], social value [29], and conversational value [28]. Our proposed sequential mediation explains the differences in the processing of external sources of information. Moreover, the findings of this study have practical implications for customer relationship management following the sale of products associated with each purchase type. Maintaining purchase-related happiness after material purchases requires more attention than experiential purchases. To achieve this, communicating with customers through email marketing or customized channels can help them to sustain their high level of commitment and selectively avoid negative information.

## 2. Literature Review and Hypothesis Development

### 2.1. Selective Information Processing

Individuals often do not pay attention to all the information they are exposed to [17,30]. As taking in every piece of information may be inefficient, they tend to process it selectively. This selective information processing is a process of encoding and processing information selectively [13]. Notably, individuals are likely to pay attention to information that is consistent with their initial attitudes, beliefs, and behaviors, in other words, to confirm their position [12,31,32]. This confirmation bias through selective information processing is often studied in the context of cognitive dissonance [33]. When individuals receive information that is contrary to their initial positions, they experience cognitive dissonance [34]. Those who experience cognitive dissonance are likely to invest their efforts to avoid an uncomfortable state [35]. One way to reduce cognitive dissonance is to focus selectively on information that is in line with their attitudes, beliefs, and behaviors [36,37].

Several researchers have explored the factors that lead to selective information processing. For example, when individuals are in a negative mood state, they tend to rely more on positive than negative information when they have a positive evaluation of the product [12]. Additionally, when individuals are highly confident in their initial beliefs, they are more likely to focus on attitude-consistent information than attitude-inconsistent information [38]. Moreover, variables affecting cognitive capacities such as information load [16,18] and commitment [19] have an influence on the type of information that individuals selectively focus on. When individuals have limited cognitive capacity due to depletion, they tend to process attitude-consistent information rather than attitude-inconsistent information [16]. Furthermore, when individuals are highly committed to their attitudes, they are more likely to select persuasive messages that are consistent with their attitudes [19]. However, there is limited literature that examines the impact that the target of decision-making has on selective information processing.

### 2.2. Purchase Type and Commitment-to-Purchase

Many studies have proposed various classifications of goods, with the dichotomy of hedonic and utilitarian goods [39] being the most widely used in the marketing literature. However, as consumer income increases, hedonic value becomes a more important criterion for choosing among potential options. This has led to the introduction of the concepts of material and experiential purchases, which differ in the source of hedonic value [9,22]. A material purchase is primarily intended to acquire material possessions, while an experiential purchase is primarily aimed at acquiring life experiences [22]. Examples of material purchases include laptop computers, cell phones, and paintings, while intangible objects such as ski trips, beach vacations, and trips abroad are examples of experiential purchases. Although the distinction between material purchases and experiential purchases was initially ambiguous, many distinguishing characteristics have been proposed by studies. First, material purchases are tangible and have more identifiable attributes, making them easier to compare to other alternatives [8]. Individuals who make material purchases are more likely to go through comparison processes when making decisions, resulting in greater decision difficulty [24]. Second, individuals are likely to use different decision strategies depending on the purchase type. Because material purchases are more comparable to other alternatives, individuals who make material purchases invest greater effort into the decision-making process and attempt to achieve the “best” option, adopting a maximizing strategy. On the other hand, individuals who make experiential purchases are likely to adopt a sufficiency strategy and settle for an option that is “good enough” due to the difficulty in comparison [8]. 

This research proposes that the reason why purchase types influence selective information processing is due to the different levels of commitment-to-purchase across purchase types. Evidence has been provided to support the relationship between purchase type and commitment-to-purchase. For example, Carter and Gilovich [8] examined the mediating effect of the present concern for the relationship between purchase type and satisfaction. Specifically, their research showed that individuals who made a material purchase had a greater present concern than those who made an experiential purchase, thus decreasing purchase satisfaction. The authors also proposed that when evaluating their purchases, individuals who made a material purchase spent more time looking for foregone alternatives than those who made an experiential purchase. As commitment-to-purchase is defined as an emotional attachment to a purchase [40], concern for making the right choice and thinking about foregone alternatives are related to the concept of commitment-to-purchase. Additionally, individuals using a maximizing strategy usually have a lower commitment to the decision than those using a satisficing strategy [41]. As mentioned earlier, because individuals making material purchases are more likely to use a maximizing strategy rather than a satisficing strategy, it can be inferred that individuals who make material purchases will have a lower commitment-to-purchase compared to their counterparts. Moreover, individuals who make experiential purchases are more likely to express their self-identity in relation to their purchases than those who made material purchases [28,42], which can be additional evidence of the relationship between commitment-to-purchase and purchase type. Recent research conducted in the context of social media has shown that online posts about different types of purchases can evoke varying perceptions of authenticity [1]. The study found that experiential purchases tend to elicit a higher level of authenticity compared to material purchases. This suggests that experiential purchases are better-suited for evaluating the relationship with consumer identity and are closely linked to commitment-to-purchase. 

Therefore, we hypothesize as follows:

**Hypothesis** **1** **(H1).**
*Individuals with experiential purchases perceive a higher level of commitment to the purchase than those with material purchases.*


### 2.3. Purchase Type and Selective Information Processing

The relationship between purchase type and selective information processing is explained in relation to the commitment-to-purchase hypothesized above. Existing literature has shown that the difference in the level of commitment influences selective information processing. In the brand commitment context, individuals with a higher level of commitment perceive positive information to be more diagnostic than negative information when evaluating a brand [43]. Moreover, in the experiment conducted by Sparks and colleagues [41], the level of commitment mediated the effect of the decision strategy on the effort to revise an initial preference. As investing effort to revise a preference is closely associated with selective information processing, their experiments support that those with a higher commitment to decisions pay attention selectively to attitude-consistent information rather than attitude-inconsistent information. Additionally, in the motivation context, individuals with a higher commitment to their initial attitudes, beliefs, and behaviors exhibit defensive motivation and a greater degree of confirmation bias. Conversely, those with lower commitment show accuracy motivation, are less likely to exhibit confirmation bias, and rely less on attitude-consistent information compared to attitude-inconsistent information [31]. A similar pattern can be found in the literature on the spillover effect. When individuals with higher commitment receive negative information that cannot be counterargued, they ignore the message and show as favorable an attitude toward the target as ever [15,44].

In this regard, it is proposed that individuals with experiential purchases will have a greater level of commitment to the purchase compared to those with material purchases. Furthermore, such differences in commitment-to-purchase will influence selective information processing; thus, experiential purchases tend to lead to a greater reliance on positive information relative to negative information. Therefore, we hypothesize that:

**Hypothesis** **2** **(H2).**
*Individuals with experiential purchases are more likely to rely on positive (vs. negative) information than those with material purchases.*


### 2.4. Purchase-Related Happiness and the Moderation of Trustworthy Feedback

Most of the previous literature on purchase type agrees that experiential purchases lead to greater and more sustainable happiness than material purchases. There are several reasons why the level of happiness over time differs across purchase types. In terms of purchase characteristics, the difference in comparability driven by the tangibility of each purchase type is one factor that influences well-being [8]. In terms of interpersonal relationships, consumers perceive greater social value [24,27] and attain greater conversational value [28] from experiential (as opposed to material) purchases. Moreover, in the post-purchase stage, experiential purchases are more easily justified by consumers themselves, even for the expensive ones [45].

**Hypothesis** **3** **(H3).**
*Individuals with experiential purchases feel greater purchase-related happiness than those with material purchases.*


To enhance the robustness of the underlying mechanism, we propose that the presence of trustworthy feedback increases the level of commitment for material purchases. Generally, individuals tend to have a higher level of commitment when they receive positive feedback [46]. Moreover, individuals who use a maximizing strategy rely more on information from external sources that are trustworthy, such as experts or family members [47]. As mentioned earlier, material purchases often lead to a higher level of concern about making the right choice [8]. Based on this, we expect that individuals who make material purchases will experience a decrease in their concern about making the right choice when they receive positive feedback. As a result, their level of commitment to the purchase will increase. We predict that the happiness generated by material purchases can be similar to that of experiential purchases, as individuals who make material purchases tend to rely more on positive information to ensure their attitude. Therefore, we hypothesize that there is a relationship between purchase type and purchase-related happiness, as follows, and visualize the research model in Figure 1. 

**Hypothesis** **4** **(H4).**
*The interactive effect of purchase type and trustworthy feedback on purchase-related happiness is sequentially mediated by commitment-to-purchase and relative reliance on positive (vs. negative) information.*


## 3. Method

### 3.1. Participants

Two hundred and twenty-three undergraduate students (male = 22.8%, female = 49.3%) participated in the experiment for course credit as a reward. They were randomly assigned to one of the four conditions in a 2 (purchase type: material vs. experiential) × 2 (feedback: present vs. not present) between-subjects design. Participants were told that the purpose of the study was to examine consumers’ purchase-related happiness.

### 3.2. Materials

The two product types were adopted from the stimuli utilized by Van Boven et al. [48]. In each condition, information regarding specific attributes of the product was provided. The results of the pretest indicated that there was no significant difference in attitude towards either product (Mwallet = 5.58, Mmusical = 5.39, F(1, 22) < 1, *p* > 0.10). In addition, the wallet was perceived as a material purchase, whereas the musical was perceived as an experiential purchase (1 = material purchase, 7 = experiential purchase, Mwallet = 2.33, Mmusical = 6.00, F(1, 22) = 44.20, *p* < 0.01). After reading the series of scenarios regarding the purchase, participants responded to measures for the following items, all of which were measured on 7-point Likert scales. Participants’ purchase-related happiness was assessed by asking them to indicate their agreement with two statements based on previous research [49]: “I feel happy with the purchase I made” and “I am pleased when I think of the purchase I made” (α = 0.841). We measured participants’ commitment-to-purchase by slightly modifying three items from Dooley and Fryxell [50]. The original items measured team members’ commitment to their decision, so we modified the wording of the scale to measure individuals’ commitments to their purchases. Participants indicated the extent of their commitment to each purchase on the following items: “I am proud to tell others about my purchase”, “I talk to others about my purchase decision”, and “I think there is much to be gained by making this purchase”. These items were then averaged to form a commitment-to-decision index. (α = 0.867). Consistent with Yoon et al. [20], reliance on positive and negative information was measured by asking participants to indicate the extent to which they relied on positive (e.g., “it is really fine leather”) and negative (e.g., “the wallet is discolored as time goes by”) commentaries on two scales, respectively (1 = not at all and 7 = very much). We then calculated a relative reliance on positive (vs. negative) information score by subtracting the negative information score from the positive. To check for purchase type manipulation, participants rated the extent to which their purchase was related to a material or experiential purchase [8]. Specifically, participants were provided with a brief explanation of the two purchase types (i.e., material and experiential purchases) and indicated whether their purchase, a wallet or a musical, was a material or an experiential purchase on a 7-point scale (1 = definitely material purchase, 7 = definitely experience purchase). Participants also indicated perceived importance of the purchase, self-assessed knowledge, credibility, amount of information, and reading enjoyment, all on 7-point scales.

### 3.3. Procedure

Participants first received the purchase type manipulation. Specifically, they were instructed to imagine that they bought a wallet and owned it (material purchase condition) or they bought a musical ticket and experienced it (experiential purchase condition). Then, participants received a manipulation for trustworthy feedback. Participants were asked to imagine that they met a friend while they were thinking about their purchase. In the condition of the presence of trustworthy feedback, participants received positive messages about their purchase from the friend who was depicted as being knowledgeable about the product. In the no-feedback condition, however, participants received only greeting messages from the friend. 

Next, participants were shown six consumer commentaries about the purchase from an Internet shopping site (or Internet ticketing site). Among six commentaries, three were positive and three were negative about the purchase. Through a pretest, the positive and negative consumer commentaries were selected. In the pretest, participants examined the extent to which each commentary contained positive or negative aspects of the wallet (or the musical) (−3 = very negative, +3 = very positive). Results of pretest showed that participants perceived fine leather, award-winning design, and appropriate size as positive attributes (M_leth_ = 2.50, M_desi_ = 2.50, M_size_ = 2.00) and discoloration, poor needlework, and fingerprint residue as negative attributes for the wallet (M_disc_ = −2.00, M_need_ = −2.33, M_fing_ = −2.28). On the other hand, participants perceived good stage setting, award-winning performance, and attractions such as tangos and tap dances as positive attributes (M_stag_ = 2.55, M_perf_ = 2.82, M_spec_ = 2.18) and long distance to the stage, uncomfortable seats, and inefficient performance hall management as negative attributes for the musical (M_fars_ = −2.18, M_seat_ = −2.73, M_unco_ = −2.18). The overall attribute importance ratings showed no significant differences between the material and experiential purchase conditions (M_material_positive_ = 2.33 vs. M_experiential_positive_ = 2.52, F(1, 68) < 1), (M_material_negative_ = −2.19 vs. M_experiential_negative_ = 2.48, F(1, 68) < 1). In each condition, six commentaries were randomly presented to participants.

Afterward, participants reported their purchase-related happiness, their commitment to the purchase, and their reliance on both positive and negative information about the purchase. Subsequently, participants completed control measures, manipulation checks, and demographic items. Finally, participants were asked about the purpose of the study, thanked, and debriefed.

## 4. Results

### 4.1. Manipulation Checks and Other Measures

An ANOVA on the perception of purchase type results revealed that participants in the material purchase condition perceived their purchase to be more material in nature than those in the experiential purchase condition did (M_material_ = 2.62 vs. M_experiential_ = 5.72; F(1, 219) = 146.33, *p* < 0.001). The control variables had no significant effect on the outcome variables. In detail, results indicated that prior knowledge about the purchased product (either wallet or musical) did not significantly vary across conditions (F(1, 219) < 1). Additionally, participants across conditions did not differ in terms of their perceived trustworthiness of commentaries provided by the internet shopping (vs. reservation) site (F(1, 219) < 1), nor did they differ in terms of reading enjoyment (F(1, 219) < 1) and perceived amount of commentaries (F(1, 219) < 1). In addition, participants in the feedback-present condition reported that the friend’s message was more relevant to their purchase than did participants in the no-feedback condition (2.76 vs. 5.48; F(1, 219) = 93.29, *p* < 0.001). Participants in the feedback-present condition rated the friend’s comment as more positive about their purchase than did participants in the no-feedback condition (4.02 vs. 5.65; F(1, 219) = 56.49, *p* < 0.001). Further analysis revealed that the valence of the friend’s message in the feedback-present condition was significantly different from the midpoint of the scale (4) in the expected direction (t(115) = 11.28, *p* < 0.001), but the valence of the message in the no-feedback condition was not significantly different from the scale midpoint (t(115) = 0.14, *p* = 0.889). Results also showed that prior knowledge about the purchased product (either wallet or musical) did not significantly differ across conditions (F(1, 219) < 1). Furthermore, participants across conditions did not differ in their perception of trustworthiness of commentaries provided by the internet shopping (vs. reservation) site (F(1, 219) < 1), nor did they differ on measures about reading enjoyment (F(1, 219) < 1) and perceived amount of commentaries (F(1, 219) < 1).

### 4.2. Hypothesis Testing

#### 4.2.1. Main Effects of Purchase Type

First, an ANOVA was conducted on the purchase-related happiness, revealing the main effect of purchase type. Consistent with Hypothesis 3, participants in the experiential purchase condition showed greater purchase-related happiness compared to those in the material purchase condition (4.36 vs. 5.38; F(1, 219) = 11.08, *p* = 0.001). To test the effect on commitment-to-purchase, another ANOVA was conducted on the commitment-to-purchase. Results showed that participants in the experiential purchase condition had greater commitment to the purchase compared to those in the material purchase condition (4.80 vs. 5.48; F(1, 219) = 12.88, *p* < 0.001), supporting Hypothesis 1. Furthermore, an ANOVA was conducted to test the relative reliance on positive versus negative information, as proposed in Hypothesis 2. Results showed that participants in the experiential purchase condition relied more on positive information compared to those in the material purchase condition (0.20 vs. 0.98; F(1, 219) = 5.83, *p* = 0.017), supporting Hypothesis 2.

#### 4.2.2. Mediation Analysis

We conducted moderated mediation analyses to test Hypothesis 4, which examines the interactive effect of purchase type and feedback presence on relative reliance on positive (vs. negative) information and purchase-related happiness. The test was performed sequentially because the process we proposed involves two mediation processes that lead to purchase-related happiness, as suggested by Muller et al. [51].

The first finding of the analysis is that commitment-to-purchase mediates the interactive effect of purchase type and positive feedback presence on purchase-related happiness (see Table 1). Following Muller et al.’s [51] study, we conducted three regression analyses. In Equation 1, the interactive effect of purchase type and feedback presence significantly affected purchase-related happiness (*p* = 0.010). In Equation 2, the interactive effect of purchase type and feedback presence also significantly affected commitment to the purchase (*p* = 0.044). In Equation 3, when the potential mediator (i.e., commitment-to-purchase) was included as a predictor, the interactive effect of purchase type and feedback presence was weakened (*p* = 0.599), and commitment-to-purchase became a significant predictor (*p* < 0.001). These results indicate that commitment-to-purchase mediates the interactive effect of purchase type and feedback presence on purchase-related happiness.

The second finding of the analysis is that the relative dependence on positive (as opposed to negative) information mediates the interactive effect of purchase type and positive feedback presence on purchase-related happiness (see Table 2). In Equation 1, the interactive effect of purchase type and feedback presence was a significant predictor of purchase-related happiness (*p* = 0.049). In Equation 2, the interactive effect of purchase type and feedback presence was also a significant predictor of the relative reliance on positive (as opposed to negative) information (*p* = 0.034). In Equation 3, when the potential mediator (i.e., relative reliance on positive (vs. negative) information) was included as a predictor, the interactive effect of purchase type and feedback presence became insignificant (*p* = 0.164), and the relative dependence on positive (vs. negative) information significantly predicted purchase-related happiness (*p* = 0.030).

The third finding of our analysis is that commitment-to-purchase acts as a mediator for the interactive effect of purchase type and positive feedback presence on relative reliance on positive (versus negative) information (see Table 3). We conducted three regression equations in a similar manner. In Equation 1, the interactive effect of purchase type and feedback presence was a significant predictor of relative reliance on positive (versus negative) information (*p* = 0.045). In Equation 2, the same interactive effect was a significant predictor of commitment-to-purchase (*p* = 0.032). However, in Equation 3, the interactive effect was not a significant predictor of relative reliance on positive (versus negative) information when including the potential mediator, commitment-to-purchase (*p* = 0.137).

These results support our hypothesis that the interactive effect of purchase type and feedback presence on purchase-related happiness is mediated sequentially by the commitment-to-purchase and relative reliance on positive (vs. negative) information.

## 5. Discussion

This research examines how experiential purchases affect selective information processing. The results indicate that individuals who make experiential purchases rely more on positive information than negative information compared to those who make material purchases. This is due to their greater commitment to the purchase. Furthermore, we demonstrate that feedback moderates purchase-related happiness. Trustworthy feedback induces individuals who make material purchases to rely more on positive information because they perceive a greater commitment-to-purchase, similar to those who make experiential purchases. The experiment provides evidence that the difference in purchase-related happiness between material and experiential purchases persists during the stage of information processing of external sources.

This research makes the following theoretical contributions to the existing body of literature. Firstly, we propose a new antecedent that leads to selective reliance on positive (vs. negative) information from the perspective of selective information processing literature. While our research is ultimately related to commitment and selective information processing, prior literature on commitment has mainly focused on commitment to attitude or commitment that arises during attitude formation [14,15]. However, the commitment that we examine in this research is triggered by the purchase type, thereby broadening the range of variables that influence selective information processing.

Secondly, from a perspective of purchase type literature, we enhance the understanding of purchase type. Previous literature on purchase type has suggested that individuals who make material purchases experience relatively lower purchase-related happiness compared to those who make experiential purchases [8,22,23,24]. However, we demonstrate that this pattern may not always occur, depending on the situations that influence the type of information individuals selectively focus on. Especially, it has been demonstrated that when material purchasers possess a high level of commitment, their happiness related to purchases can be sustained positively through selective information processing.

Thirdly, previous studies have examined the superiority of experiential purchases by examining the distinct characteristics associated with each purchase type. However, this study identifies the differences in information processing that may arise between the two types of purchases. Therefore, this research contributes to the existing literature by introducing a new underlying process between purchase type and purchase-related happiness through the sequential mediation of commitment-to-purchase and relative reliance on positive (versus negative) information.

This study provides an important managerial implication for building and managing customers’ happiness related to their purchases. To achieve sustainable benefits, companies must prioritize managing customers before they make purchases. Specifically, this research suggests that companies dealing with material products should pay greater attention to customers, as they have lower commitment to their purchases and are more sensitive to negative commentaries than positive ones. Based on our findings, managers should enhance purchase-related happiness by increasing customers’ commitment to their purchases through various CRM tactics. One way to improve commitment is to provide customers with signals, such as evaluations from authorized institutions, to reassure them that their decisions are correct. 

The results of this study suggest that the source of information that consumers rely on to determine their happiness in the post-purchase stage varies. Depending on whether the product is positioned as an experience or a possession, the communication channel to focus on may differ. For experiential purchases, an online review with a more vivid media template should be constructed if there are many positive reviews. Conversely, for material purchases, visualization should be used to reduce exposure to the opinions of other consumers. This can be achieved through the analysis of consumer logs for those who have already made a purchase.

## 6. Limitations and Further Research

In the experiment, we manipulated the purchase type, material and experiential purchases, with the pretested products. However, participants might not have perceived the purchase as we intended. For the material purchase, for example, individuals might not have recognized that they will “possess” the leather wallet. They might have felt that they would “experience” the wallet that we showed in the experiment. Similarly, for the experiential purchase, participants might not have thought that they would experience the musical. Experiential purchases often exist only in the memory, but individuals in the experiential condition only got to know about the musical ticket through the experiment. Even though the manipulation check results showed that the participants classified the wallet as a material purchase and the musical as an experiential purchase, this might not fully reflect how individuals perceived each purchase type. Moreover, greater reliance on positive information, as exhibited by the participants, may be a result specific to the musical. For example, moviegoers also consider negative information extensively when they make decisions on what to watch. The findings of this study regarding the relationship between purchase-related happiness and purchase type align with previous research, indicating their generalizability. Nonetheless, additional confirmation is necessary to establish the mediating effect of selective information processing. Thus, further research is necessary to examine whether our prediction holds for various product types to add robustness to our predictions regarding the effect of the purchase type in real purchase situations. For instance, manipulation methods such as positioning identical purchases as either material or experiential purchases are expected to enable control of the effects that products such as musical or wallets can create [52]. 

Additionally, we did not consider the possible effect of differences in information loads across purchase types. As mentioned earlier, individuals who make a material purchase are more likely to use a maximizing rather than satisficing strategy. Because individuals using a maximizing strategy exert greater effort to interpret given information than individuals using satisficing strategies, the perceived number of six commentaries could be smaller for maximizers than satisficers [53,54]. If the perceived information load differs across purchase types, then the selective processing of positive (vs. negative) information can also differ; individuals in the material purchase condition perceive that there is relatively less information available, thus relying more on negative than positive information. However, in this study, we ruled out this alternative explanation by conducting an ANOVA on the perceived information load (3.04 vs. 2.96; F(1, 222) = 0.12, *p* > 0.10). In other words, purchase type did not influence the perceived information load. 

In this research, we used the dichotomy of material and experiential purchases to represent the purchase target. Future research should explore other variables that influence selective information processing. For example, for identical products, information processing can differ between a group of first-time purchasers and a group of repeated purchasers. The repetition might increase the sensitivity to cognitive dissonance, so the repeated purchasers would give a greater effort to reducing the dissonance through information processing. The exploration of moderators for the effect of purchase type on the commitment-to-purchase is another research direction that can be considered. For example, if the personal connection becomes greater for the material purchase, then an effect that is similar to that of positive feedback from an external source can emerge because material purchases, in general, offer a less personal connection than experiential purchases [55]. Increasing the degree of personal connection will increase the level of commitment individuals experience regarding the purchase. 

In future research, more extensive investigation of the characteristics of purchase type is necessary. In this research, we mainly focused on the differences in comparability and decision strategies to examine the effect of purchase type on the commitment-to-purchase. However, many differences have been found between the two purchase types, which may even have confounding effects. Thus, it is necessary to manipulate and examine each characteristic separately for a deeper investigation of the underlying process. Such dissemination of characteristics is also necessary for decision strategies in the future since it also involves various characteristics that distinguish each strategy type.

## Figures and Tables

**Figure 1 behavsci-13-00396-f001:**
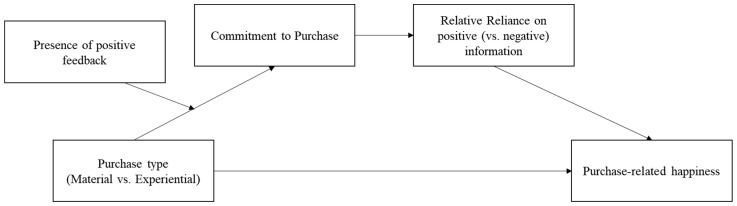
Research model.

**Table 1 behavsci-13-00396-t001:** Summary of regressions for moderated mediation test (1).

	Equation 1: Purchase-Related Happiness			Equation 2: Commitment-to-Purchase			Equation 3: Purchase-Related Happiness		
Predictor	B	SE B	*β*	B	SE B	*β*	B	SE B	*β*
X, Purchase type	0.29	0.11	0.26 ***	0.35	0.10	0.33 ***	0.00	0.08	0.00
Mo, Feedback presence	0.27	0.11	0.24 **	0.36	0.10	0.34 ***	−0.06	0.42	−0.06
XMo Interaction	−0.22	0.11	**−0.20 ****	−0.21	0.10	**−0.20 ****	−0.41	0.08	−0.04 ^a^
Me, Commitment-to-purchase							−0.83	0.08	**0.80 *****
MeMo Interaction							0.01	0.08	0.02

Note: Bold indicates that the values of *β* are required to be significant to prove moderated mediation. The values of *β*
^a^ are required to be insignificant to prove a fully moderated mediation. ** *p* < 0.05, *** *p* < 0.01.

**Table 2 behavsci-13-00396-t002:** Summary of regressions for moderated mediation test (2).

	Equation 1: Purchase-Related Happiness			Equation 2: Relative Reliance			Equation 3: Purchase-Related Happiness		
Predictor	B	SE B	*β*	B	SE B	*β*	B	SE B	*β*
X, Purchase type	0.29	0.11	0.26 ***	0.40	0.17	0.24 **	0.23	0.11	0.21 **
Mo, Feedback presence	0.27	0.11	0.24 **	0.46	0.17	0.27 ***	0.20	0.12	0.18
XMo Interaction	−0.22	0.11	**−0.20 ****	−0.36	0.17	**−0.21 ****	−0.16	0.11	−0.15 ^a^
Me, Relative reliance							0.15	0.07	**0.24 ****
MeMo Interaction							−0.00	0.07	−0.01

Note: Bold indicates that the values of *β* are required to be significant to prove moderated mediation. The values of *β*
^a^ are required to be insignificant to prove a fully moderated mediation. ** *p* < 0.05, *** *p* < 0.01.

**Table 3 behavsci-13-00396-t003:** Summary of regressions for moderated mediation test (3).

	Equation 1: Relative Reliance			Equation 2: Commitment-to-Purchase			Equation 3: Relative Reliance		
Predictor	B	SE B	*β*	B	SE B	*β*	B	SE B	*β*
X, Purchase type	0.40	0.17	0.24 **	0.35	0.10	0.33 ***	0.22	0.17	0.13
Mo, Feedback presence	0.46	0.17	0.27 ***	0.36	0.10	0.34 ***	−0.06	0.91	−0.04
XMo Interaction	−0.36	0.17	**−0.21 ****	−0.21	0.10	**−0.20 ****	−0.26	0.17	−0.15 ^a^
Me, Commitment-to-purchase							0.56	0.17	**0.35 *****
MeMo Interaction							0.06	0.17	0.19

Note: Bold indicates that the values of *β* are required to be significant to prove moderated mediation. The values of *β*
^a^ are required to be insignificant to prove a fully moderated mediation. ** *p* < 0.05, *** *p* < 0.01.

## Data Availability

The data of this study are available from the corresponding author upon reasonable request.

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
