# Peer review of "The Influence of Consumer Purchases on Purchase-Related Happiness: A Serial Mediation of Commitment and Selective Information Processing"

_behavsci, 2023, doi:10.3390/bs13050396_

Round 1
Reviewer 1 Report
It could be great, if authors use more newest sources of the information. Most of sources in the reference list are more than 10 years old.
Author Response
Response to Reviewer 1:
- It could be great, if authors use more newest sources of the information. Most of sources in the reference list are more than 10 years old.
à Thank you for providing us with your valuable feedback. Based on your suggestion, we have included more recent references and updated the relatively outdated ones. The references that have been newly added are listed below.
Dai, H., Chan, C., & Mogilner, C. (2020). People rely less on consumer reviews for experiential than material purchases. Journal of Consumer Research, 46(6), 1052-1075.
Gilovich, T., & Gallo, I. (2020). Consumers’ pursuit of material and experiential purchases: A review. Consumer Psychology Review, 3(1), 20-33. https://doi.org/10.1002/arcp.1053
Hart, W., Richardson, K., Tortoriello, G. K., & Earl, A. (2020). ‘You Are What You Read:’Is selective exposure a way people tell us who they are?. British Journal of Psychology, 111(3), 417-442.
Jabr, W., & Rahman, M. S. (2022). ONLINE REVIEWS AND INFORMATION OVERLOAD: THE ROLE OF SELECTIVE, PARSIMONIOUS, AND CONCORDANT TOP REVIEWS. MIS Quarterly, 46(3).
Momsen, K., & Ohndorf, M. (2022). Information avoidance, selective exposure, and fake (?) news: Theory and experimental evidence on green consumption. Journal of Economic Psychology, 88, 102457.
Morewedge, C. K., Monga, A., Palmatier, R. W., Shu, S. B., & Small, D. A. (2021). Evolution of consumption: A psychological ownership framework. Journal of Marketing, 85(1), 196-218.
Pizzutti, C., Gonçalves, R., & Ferreira, M. (2022). Information search behavior at the post-purchase stage of the customer journey. Journal of the Academy of Marketing Science, 50(5), 981-1010. https://doi.org/10.1007/s11747-022-00864-9
Tezer, A., & Bodur, H. O. (2020). The greenconsumption effect: How using green products improves consumption experience. Journal of Consumer Research, 47(1), 25-39.

Reviewer 2 Report
I’m glad to have the opportunity to review the paper. Overall, the paper is well-crafted and interesting. Congratulations to the authors.
I have just a few comments/notes.
1 - Line 225
It is not “2. Method”, but “3. Method”.
2 - Lines 257-8
The authors mention “Subsequently, participants completed control measures, manipulation 258 checks, and demographic items.” One would expect that they would provide details about that… or are those details provided in 3.1 (line 265) and 4.1 (line 295)?
By the way, when in line 261 we have “3.1. Measure”, one would expect to encounter 3.2, 3.3… Otherwise, why not move from point 3 (line 225) to point 4 (line 294)? There is a problem with the numbering/allocation on pages 6 and 7.
3 - Lines 280, 370, and 488 – Please check the spaces…
4 - Unlikely the previous notes, there is a final and more profound note/concern:
The limitations and future research, in the end, wisely resolve most of the reservations and notes that the document refers to; however, without prejudice to better judgment, some questions can be raised: (i) the context, taken into account in previous studies, can it be ignored?; (ii) the participants are who they are and not others (different background and/or different age group and/or different culture, etc. - note that the study does not mention the continent/country/region in which the study was carried out) no limits the generalizability of the results?; (iii) the experiences and stimuli created, having their merits, to what extent are they not also limitations in themselves, subject to reference at the end of the article (point 6)?
All the best to the authors, regarding this and other research.
Author Response
Response to Reviewer 2:
I’m glad to have the opportunity to review the paper. Overall, the paper is well-crafted and interesting. Congratulations to the authors.
I have just a few comments/notes.
1 - Line 225
It is not “2. Method”, but “3. Method”.
à Thank you for your comments. I have revised the error in the heading.
2 - Lines 257-8
The authors mention “Subsequently, participants completed control measures, manipulation 258 checks, and demographic items.” One would expect that they would provide details about that… or are those details provided in 3.1 (line 265) and 4.1 (line 295)?
à Thank you for your valuable comments. As per your suggestion, it would be appropriate to incorporate sample demographic information. However, in this study, we only gathered age and gender information to limit the disclosure of personal data. Consequently, we have included gender information in section 3.1 of the sample, but we have not created a separate table for it.
By the way, when in line 261 we have “3.1. Measure”, one would expect to encounter 3.2, 3.3… Otherwise, why not move from point 3 (line 225) to point 4 (line 294)? There is a problem with the numbering/allocation on pages 6 and 7.
à Thank you for your valuable comments. As your suggestion, there were some issues with numbering and allocation. We have rectified the problem by renumbering "3.1. Measure" to "3.2 Materials" and labeling the preceding content as "3.1 Participants." and “3.3 Procedure”
3 - Lines 280, 370, and 488 – Please check the spaces…
à Thank you for your comments. Based on your comment, We conducted a thorough proofreading to identify any errors in the use of spacing. After making the necessary corrections, I can confirm that there are no grammatical errors in the Word file. Please inform me if any additional modifications are required.
4 - Unlikely the previous notes, there is a final and more profound note/concern:
The limitations and future research, in the end, wisely resolve most of the reservations and notes that the document refers to; however, without prejudice to better judgment, some questions can be raised: (i) the context, taken into account in previous studies, can it be ignored?; (ii) the participants are who they are and not others (different background and/or different age group and/or different culture, etc. - note that the study does not mention the continent/country/region in which the study was carried out) no limits the generalizability of the results?; (iii) the experiences and stimuli created, having their merits, to what extent are they not also limitations in themselves, subject to reference at the end of the article (point 6)?
à Thank you for providing a detailed review of the limitations and proposed future research in this study.
(i) This study employs a definition of material and experiential purchases that is consistent with previous research and utilizes a similar manipulation to that used in prior studies. As a result, the experiments in this study were conducted in environments that are similar to those considered in previous research.
(ii) The experiment for this study was carried out on undergraduate students in South Korea. The main effect discovered in this study has been consistently found in all studies that focus on material and experiential purchases. Therefore, it is expected that the specificity of the experimental environment will not impede the generalizability of the study results. However, as you have pointed out, the mediation effect on selective information processing needs to be replicated in a broader range of samples. This limitation has been acknowledged in the study.
Page 11.
The findings of this study regarding the relationship between purchase-related happiness and purchase type align with previous research, indicating their generalizability. Nonetheless, additional confirmation is necessary to establish the mediating effect of selective information processing.
(iii) As mentioned previously in the limitations of this study, participants perceived their stimuli as individual purchases. However, it is unclear whether they perceived them as actual purchases. To overcome this limitation, we recommend using an alternative manipulation method and suggest it for future research. Please refer to the following section for more details.
Page 11.
For instance, manipulation methods like positioning identical purchases as either material or experiential purchases are expected to enable the control of the effects that products such as musical or wallets can create (Goodman et al., 2019).

Reviewer 3 Report
This study has provided some interesting findings. However, the authors are recommended to provide some statistics, examples (real-world examples) and recent developments in the introduction section to justify the importance of this study.
In addition, the authors are also recommended to highlight the contributions clearly in the introduction section, and discussion section.
Importantly, The authors have mentioned the importance of "review" in the paper. However, social media influencers' endorsement and review is very important. Thus, the authors are recommended to cite the following papers to improve the quality of this study:
Cheung, M. L., Leung, W. K., Aw, E. C. X., & Koay, K. Y. (2022). “I follow what you post!”: The role of social media influencers’ content characteristics in consumers' online brand-related activities (COBRAs). Journal of Retailing and Consumer Services, 66, 102940.
Jin, S. V., Muqaddam, A., & Ryu, E. (2019). Instafamous and social media influencer marketing. Marketing Intelligence & Planning, 37(5), 567-579.
Vrontis, D., Makrides, A., Christofi, M., & Thrassou, A. (2021). Social media influencer marketing: A systematic review, integrative framework and future research agenda. International Journal of Consumer Studies, 45(4), 617-644.
Author Response
Response to Reviewer 3:
- This study has provided some interesting findings. However, the authors are recommended to provide some statistics, examples (real-world examples) and recent developments in the introduction section to justify the importance of this study.
à Thank you for your comments. As your suggestion, we have included a real-world example. Please see below.
Page 1.
For example, over 92 percent of consumers consult product reviews prior to making a purchase decision, and these reviews serve as a dependable sales assistant, contributing to the growth in sales revenue (Spivack, 2019; Le et al., 2022).
Based on the analysis of data collected from product evaluation forums, it was discovered that 38.8% of reviews were acquired after the purchase, rather than prior to it (Pizzutti et al., 2022).
- In addition, the authors are also recommended to highlight the contributions clearly in the introduction section, and discussion section.
à Thank you for your comments. As you mentioned, I have revised the introduction and discussion sections. Specifically, I have included a discussion on communication methods to enhance the practical implications section.
Page 2.
Moreover, the findings of this study have practical implications for customer relationship management following the sale of products associated with each purchase type. Maintaining purchase-related happiness after material purchases requires more attention than experiential purchases. To achieve this, communicating with customers through email marketing or customized channels can help sustain their high level of commitment and selectively avoid negative information.
Page 10.
This research examines how experiential purchases affect selective information pro-cessing. The results indicate that individuals who make experiential purchases rely more on positive information than negative information compared to those who make material purchases. This is due to their greater commitment to the purchase. Furthermore, we demonstrate that feedback moderates purchase-related happiness. Trustworthy feedback induces individuals who make material purchases to rely more on positive information because they perceive a greater commitment to purchase, similar to those who make expe-riential purchases. The experiment provides evidence that the difference in purchase-related happiness between material and experiential purchases persists during the stage of information processing of external sources.
This research makes the following theoretical contributions to the existing body of lit-erature. Firstly, we propose a new antecedent that leads to selective reliance on positive (vs. negative) information from the perspective of selective information processing literature. While our research is ultimately related to commitment and selective information pro-cessing, prior literature on commitment has mainly focused on commitment to attitude or commitment that arises during attitude formation (Hart et al., 2020; Hiemisch et al., 2002). However, the commitment that we examine in this research is triggered by the purchase type, thereby broadening the range of variables that influence selective information pro-cessing.
Secondly, from a perspective of purchase type literature, we enhance the understand-ing of purchase type. Previous literature on purchase type has suggested that individuals who make material purchases experience relatively lower purchase-related happiness compared to those who make experiential purchases (Dai et al., 2020; Van Boven & Gilo-vich, 2003; Nicolao et al., 2009; Carter & Gilovich, 2010). However, we demonstrate that this pattern may not always occur, depending on the situations that influence the type of information individuals selectively focus on. Especially, it has been demonstrated that when material purchasers possess a high level of commitment, their happiness related to purchases can be sustained positively through selective information processing.
Thirdly, previous studies have examined the superiority of experiential purchases by examining the distinct characteristics associated with purchase type. However, this study identifies the differences in information processing that may arise between the two types of purchases. Therefore, this research contributes to the existing literature by introducing a new underlying process between purchase type and purchase-related happiness through the sequential mediation of commitment to purchase and relative reliance on positive (versus negative) information.
This study provides an important managerial implication for building and manag-ing customers’ happiness related to their purchases. To achieve sustainable benefits, companies must prioritize managing customers before they make purchases. Specifically, this research suggests that companies dealing with material products should pay greater attention to customers, as they have lower commitment to their purchases and are more sensitive to negative commentaries than positive ones. Based on our findings, managers should enhance purchase-related happiness by increasing customers’ commitment to their purchases through various CRM tactics. One way to improve commitment is to pro-vide customers with signals, such as evaluations from authorized institutions, to reassure them that their decisions are correct.
The results of this study suggest that the source of information that consumers rely on to determine their happiness in the post-purchase stage varies. Depending on whether the product is positioned as an experience or a possession, the communication channel to focus on may differ. For experiential purchases, an online review with a more vivid media template should be constructed if there are many positive reviews. Conversely, for materi-al purchases, visualization should be used to reduce exposure to the opinions of other consumers. This can be achieved through the analysis of consumer logs for those who have already made a purchase.
- Importantly, The authors have mentioned the importance of "review" in the paper. However, social media influencers' endorsement and review is very important. Thus, the authors are recommended to cite the following papers to improve the quality of this study:
à Thank you for your recommendations. We have cited the recommended research in the manuscript.
Cheung, M. L., Leung, W. K., Aw, E. C. X., & Koay, K. Y. (2022). “I follow what you post!”: The role of social media influencers’ content characteristics in consumers' online brand-related activities (COBRAs). Journal of Retailing and Consumer Services, 66, 102940.
Jin, S. V., Muqaddam, A., & Ryu, E. (2019). Instafamous and social media influencer marketing. Marketing Intelligence & Planning, 37(5), 567-579.
Vrontis, D., Makrides, A., Christofi, M., & Thrassou, A. (2021). Social media influencer marketing: A systematic review, integrative framework and future research agenda. International Journal of Consumer Studies, 45(4), 617-644.

Reviewer 4 Report
Dear Authors
After reading the paper carefully, I concluded that the paper is interesting but needs improvement at some points. The research background section should present a more robust theoretical and empirical ground for constructing hypotheses. In the method section, some parts are superficial:
229-230 - Providing participants with cues on study hypotheses can introduce demand characteristics and jeopardize the validity of the results.
* 231-233 - Employing hypothetical scenarios in manipulations generally lacks ecological validity and may fall short of eliciting actual behavior.
* Method: The method section should be restructured. Please provide information on your participants first, then on materials (including the manipulations and measures), and finally, the procedure, all in separate subsections.
* Results: Please provide exact values when reporting p-values (except for those smaller than .01), instead of using thresholds.
* Limitations: The main limitation of the experiment is the use of hypothetical scenarios and the revelation of critical information about your hypotheses. Please acknowledge these issues in the limitations section.
Kind Regards,
Author Response
Response to Reviewer 4:
After reading the paper carefully, I concluded that the paper is interesting but needs improvement at some points. The research background section should present a more robust theoretical and empirical ground for constructing hypotheses. In the method section, some parts are superficial:
à Thank you for your valuable comments. During the development of hypotheses, it was discovered that there was a lack of clear reasoning regarding the connection between purchase type and commitment to purchase. To rectify this issue, the authors improved the literature review and hypothesis development section. Furthermore, the results that were only briefly mentioned in the method section were revised in accordance with the given instructions.
229-230 - Providing participants with cues on study hypotheses can introduce demand characteristics and jeopardize the validity of the results.
à In this study, we only mentioned the purpose of research is about the “happiness related to purchases” and did not specify the type of purchase or the presence of feedback. As a result, we are confident that the issue you raised would not have influenced consumers' responses.
* 231-233 - Employing hypothetical scenarios in manipulations generally lacks ecological validity and may fall short of eliciting actual behavior.
à As you mentioned, the use of hypothetical scenarios may compromise the validity of our reviews. Nevertheless, we have concluded that assuming hypothetical scenarios is the most appropriate approach to provide reviews that are relevant for consumers' purchasing decisions. In the future, we could explore the possibility of selectively processing information for identical reviews by presenting the same product as either a material or experiential good to consumers. We have included this suggestion in the limitations and future research section. Please see below.
Page 11.
The findings of this study regarding the relationship between purchase-related happiness and purchase type align with previous research, indicating their generalizability. Nonetheless, additional confirmation is necessary to establish the mediating effect of selective information processing. Thus, further research is necessary to examine whether our pre-diction holds for various product types to add robustness to our predictions regarding the effect of purchase type in real purchase situations. For instance, manipulation methods like positioning identical purchases as either material or experiential purchases are expected to enable the control of the effects that products such as musical or wallets can create (Goodman et al., 2019).
* Method: The method section should be restructured. Please provide information on your participants first, then on materials (including the manipulations and measures), and finally, the procedure, all in separate subsections.
à Thank you for your valuable comments. As you suggested, we have restructured the contents of Method
* Results: Please provide exact values when reporting p-values (except for those smaller than .01), instead of using thresholds.
à Thank you for your valuable comments. As you pointed out, we have provided the exact p-values.
* Limitations: The main limitation of the experiment is the use of hypothetical scenarios and the revelation of critical information about your hypotheses. Please acknowledge these issues in the limitations section.
à Thank you. As previously stated, we have included a limitation section addressing the hypothetical scenario. Please see below.
Page 11.
The findings of this study regarding the relationship between purchase-related happiness and purchase type align with previous research, indicating their generalizability. Nonetheless, additional confirmation is necessary to establish the mediating effect of selective information processing. Thus, further research is necessary to examine whether our pre-diction holds for various product types to add robustness to our predictions regarding the effect of purchase type in real purchase situations. For instance, manipulation methods like positioning identical purchases as either material or experiential purchases are expected to enable the control of the effects that products such as musical or wallets can create (Goodman et al., 2019).

Round 2
Reviewer 3 Report
I recommend to accept the paper in the present form.